# Buy Now Pay Later—A Fad or a Reality? A Perspective on Electronic Commerce

Dana Adriana Lupșa-Tătaru, Eliza Nichifor *, Lavinia Dovleac, Ioana Bianca Chițu, Raluca Dania Todor and Gabriel Brătucu

Faculty of Economic Sciences and Business Administration, Transilvania University of Brașov, Colina Universității Street, No. 1, Building A, 500068 Brașov, Romania; lupsad@unitbv.ro (D.A.L.-T.); lavinia.dovleac@unitbv.ro (L.D.); ioana.chitu@unitbv.ro (I.B.C.); raluca.todor@unitbv.ro (R.D.T.); gabriel.bratucu@unitbv.ro (G.B.)
* Correspondence: eliza.nichifor@unitbv.ro

**Abstract:** The Millennials and Generation Z use online shopping for a holistic experience and buy more expensive or better-quality products with buy now pay later payment methods for their highly demanding needs. The authors aimed to deepen understanding of this phenomenon by finding related knowledge fields and discovering the type of economy that will represent an increasing market share for the method of domestic e-commerce payments. The methodology used combined computer-assisted review, descriptive statistics, and linear regression to explain the market share of 23 economies worldwide. Student credit card use, myopic consumer law, buying tendencies. and dark financial triangles were identified as related topics. Logistics performance, ease of doing business, and postal development were found to be significant factors. Finally, economies with medium ranks are inclined to adopt this kind of payment easily. Hence, major implications, both managerial and academic, must be addressed. High responsibility should be borne by industry associations, which should run information campaigns by collaborating with public institutions. From the point of view of theoretical implications, studying the buy now and pay later concept and its outcomes might deepen understanding of consumer behaviour, decision-making processes, risk perception mitigation, debt behaviours, and credit adoption.

**Keywords:** buy now pay later; e-commerce; fintech; Generation Z; bibliometric analysis

## 1. Introduction

Buy-now-pay-later (hereinafter BNPL) arrangements, one of Fintech's innovative instruments, are based on 'responsible spending' and are a less expensive alternative to credit cards (Gerrans et al. 2022). They are interest-free instalment loans, an agreement between buyer and merchant to promote financing at the point of sale, but they are not yet properly regulated to protect the buyer (Soni 2023).

Advanced technology, e-wallets, e-money, and smartphone accessibility have led to changes in behaviour, and consumers are becoming more "consumptive", especially Generation Z (digital natives), constituting an opportunity to use BNPL services (Lia and Natswa 2021).

Fintech (financial technologies) represents an emerging concept in the industry of banking, offering alternative solutions to specific instruments in this industry (Vijai 2019), namely bank products and services using advanced technologies (Knewtson and Rosenbaum 2020). Using these instruments is essential to consumer retention (Tripathy and Jain 2020), increasing the number of payment options at retailers (Kliber et al. 2021). Fintech led to the digitalisation of the bank business model from "brick-and-mortar" banking (offline presence) to an online presence and essentially modified the way in which financial services are delivered (Agarwal and Zhang 2020). This concept emerged because of the very rapid evolution of technology in recent years, the use of cryptocurrencies, and the digitalisation of business models, which revolutionised mobile payment (Hendershott et al.

2021). Additionally, global financial crises and the Covid-19 pandemic period determined the need to innovate new solutions for the problems generated by these factors (Imerman and Fabozzi 2020). The Fintech industry underwent many changes in recent years because of the emergence and development of new instruments, such as blockchain, artificial intelligence, and cloud computing (Nelaturu et al. 2022). Moreover "fintech is struggling to find a sustainable business model in the face of funding instability" (World Economic Forum 2017).

Fintech's benefits include a cost economy, increased efficiency, greater transparency, increased consumer satisfaction, improvement of infrastructure components, and easier access to the financial system. Additionally, Fintech may have negative consequences related to cyber security, for example, and it requires the creation of an appropriate legislative framework (Hasan 2023; Moosa 2022). To increase the competitiveness of the EU's financial sector, in September 2020, the European Commission adopted a document that includes a legislative and financial strategy to ensure that European citizens have access to innovative financial products, as well as financial protection and stability, based on the premise that digital financial services contribute to the modernisation of the European economy and will transform it "into a global digital player" (European Commission 2023). Specialists have raised the problem of consumers' understanding of increasingly complex financial concepts that may exceed their capacity, which is why the social side must be considered in the regulation of these services to make them more useful and socially sustainable (Johnson et al. 2021). BNPL payment methods represent "FinTech credit products", whereby consumers can defer payment on purchases made through an interest-free instalment scheme. The use of this system via credit cards is more often used by young consumers and those located in more economically deprived areas (Guttman-Kenney et al. 2023).

The aim of the study is to elaborate a line of thought about buy now pay later payment methods and to gather information to answer three research questions. Due to complexity of the workflow applied to perform the study in the first place, the authors introduce a graphical representation of the researched path (Figure 1).

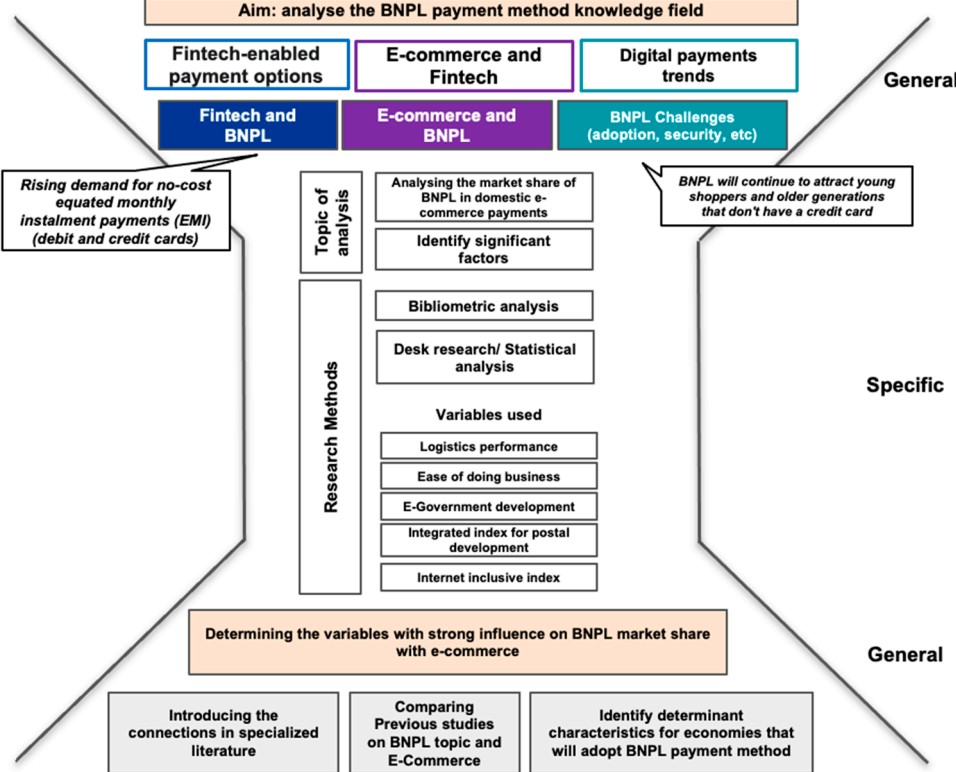

**Figure 1.** The workflow of the study.

The workflow follows a logical line by aiming to respond to three research questions. (1) How has the BNPL concept been presented in the literature? (2) What influences the market share of BNPL in domestic e-commerce payments? and (3) Who will adopt BNPL payment methods in the future? By attempting to discover the arguments for each research questions, four objectives were formulated. The authors intended to (O1) describe the core significance of BNPL payment methods in science, (O2) identify the major areas and related topics in the specialised literature, (O3) find significant variables to explain the market share of BNPL in domestic e-commerce payments, and (O4) discover determinant characteristics of economies that will adopt BNPL payment methods in the future.

Outstanding results were identified, including that inclusive internet indices or e-government development are not significant factors for BNPL payment method adoption.

All the outcomes are logically presented in the following sections. The paper starts with a literature review, which introduces the context and the knowledge field to which the BNPL concept belongs. Therefore, the methodology shows all the methods applied to obtain the results illustrated in the third section. The paper continues with discussions in the fourth part and ends with conclusions that highlight the scientific contributions, theoretical and managerial implications, and limitations of the study.

## 2. Literature Review

### 2.1. E-Commerce and BNPL

The e-commerce environment has expanded over the last decade, as users are able to access the internet through an increasing number of devices (Chung et al. 2018; Gañac 2018; Ono et al. 2012). Because of the increased ownership, usage, and portability of devices (Gañac 2018), people are spending more time online; consequently, e-commerce and mobile-commerce retail platforms have developed and become available to consumers (Park and Hill 2018). Since online shopping was normalised in most markets (Park et al. 2012), the perception of reliability and trust on the retail websites has highly increased (Clemons et al. 2016). More resources than ever have been directed towards making online retail channels evermore functional to compete with, complement, or at times replace entirely the physical store (Thaichon et al. 2018). Speed, accessibility, and ease of browsing have made e-commerce a norm rather than an exception (Gañac 2018; Ono et al. 2012; Park and Hill 2018; Park et al. 2012). When discussing e-commerce, there are some trends manifesting (Aite Novarica 2021; Bolu 2021; Di Maggio et al. 2022; McKinsey & Company 2020, 2021; Money & Pension Services 2023):

- BNPL is embedded in the shopping journey and is promoted across the payment journey right from the product pages, not only at checkout;
- Consumers are starting their shopping journey at BNPL provider apps/websites;
- Leading players have developed strong consumer brands and apps, driving incremental sales through affiliate marketing channels;
- Leading BNPL players aspire to build "super apps" that offer shopping, payments, financing, and banking products in a single platform; BNPL is becoming more available for a wider range of purchases, including everyday purchases;
- Most of BNPL users are younger than 40 years old, but increased demand is noted for all user groups;
- Notably, repayment reminders offered by certain BNPL providers have garnered remarkable support among users;
- The surge in BNPL adoption can be attributed to three essential factors: awareness, accessibility, and affordability;
- Implementing appropriate regulations and employing awareness about behavioural impulses have shown promise in promoting responsible credit management;
- However, it is essential to acknowledge that BNPL products' lack of understanding and the limited transparency in individuals' credit profiles may potentially jeopardise their financial wellbeing, with broader implications for the economy overall.

The market has already widely adopted BNPL, and Millennial and Generation Z consumers are more willing to utilise buy now pay later services to help manage the growing costs of goods, according to a new survey from Trustpilot (Trustpilot 2022). According to studies from 2021 and 2022 (Backman 2022; CUToday 2021), 62% of BNPL users (of whom 62% are between 18 and 24 years old) think it could replace their credit cards; 53% of consumers who have never used BNPL say they are at least somewhat likely to use it within the next year; 48% of BNPL users say they used it for electronics purchases—the largest usage of any industry represented; 50% more consumers say they understand BNPL compared to last year; and 48% more consumers used a BNPL service this year compared to the previous year.

Moreover, building on the e-commerce momentum from the Covid-19 pandemic, the BNPL model is one of the fastest-growing segments in consumer finance. Considering the impact of inflation and weaker consumer spending on company valuations post-pandemic, the growth potential of BNPL is likely to help it weather economic headwinds (Perlin 2021). This new trend on e-commerce sites in increasing in popularity since credit cards are declining in popularity. Millennials and Generation Z avoid credit cards and embrace e-commerce, and the BNPL market is expected to grow exponentially, keeping up with what is popular and what will earn the most sales. BNPL can stimulate e-commerce demand because it reduces customers' cash flow pressure, making it easier for potential customers who cannot afford what they want.

Aside from providing customers with the opportunity to obtain digital products sooner, using BNPL—a simple financing option—can enhance the trustworthiness of online brands, especially for young generations. According to the same survey (Backman 2022), 41% of younger generations said that it is probable they will use BNPL services for household purchases.

When discussing benefits, e-commerce may benefit from the use of BNPL services, as the conversion rate increases on average by 20–30% according to statistics (CUToday 2021; Perlin 2021). Moreover, 60% of online shoppers have already used BNPL at least once in the past year; thus, e-shops may benefit from the use of BNPL because of the reduction in the abandoned shopping carts.

BNPL thus has increased cart value, the conversion rates, and the purchase frequency, and it demonstrates a great customer experience, reduces abandoned conversion rates, increases the trustworthiness of e-shops, and offers customers more purchasing power.

BNPL enables more online shoppers to complete purchases that they perhaps could not have completed otherwise. A recent study (Backman 2022) confirmed this theory by finding that 45% of BNPL users opt for this solution to buy items that do not fit their budget. Thus, the more visitors that an online shop attracts to its site, the higher that the chance is of converting them into paying customers.

E-commerce may benefit from the use of BNPL services, as the conversion rate increases on average by 20–30%, according to statistics (CUToday 2021; Perlin 2021). If a buyer has been dissuaded from purchasing because the merchant did not offer a payment method, offering multiple payment methods, including a BNPL option, will reduce the probability of this event. Moreover, customers are more likely to purchase even sooner if they feel comfortable with their ability to pay, thus avoiding the possibility of them changing their minds, forgetting about the basket, or finding similar products from competition.

Moreover, 60% of online shoppers have already used BNPL at least once in the past year; thus, e-shops may benefit from the use of BNPL because of the reduction in abandoned shopping carts. BNPL helps with the object of providing multiple methods to pay when a product in the cart is too expensive, giving purchasers the chance to pay rates that are more affordable over time.

### 2.2. Millennials, Generation Z and Market Growth

If we refer to Millennials and Generation Z, they use online shopping and other channels for a holistic experience; buy more expensive, better-quality products or preowned

products—since they desire brands with the mission of doing good for society; and make small purchases more often (Marketing Evolution 2019; McKinsey & Company 2020; The KEENFOLKS 2023; Trustpilot 2018; Vogue Business 2021). By adopting new payment methods such as BNPL, both online and in store, adapting to new shopping habits means adjusting to different needs and different expectations, such as those of Generation Z. Additionally, BNPL responds to the needs of Millennials and Generation Z for transparency and authenticity since there are no hidden costs; thus, they are more likely not only to buy but also to become loyal in the long run.

Unhealthy financial habits, lack of trust in financial institutions, and historical financial inequities currently inhibit mass adoption and expansion of BNPL: "The existing design of BNPL offerings may not be sustainable." BNPL's welfare effects are unclear. People less informed about how to use this technique are predisposed to make naive mistakes that can damage both their financial state and their state of mind (Guttman-Kenney et al. 2023).

In 2022, 40% of customers using BNPL options fell behind on payments, and 68% of BNPL customers admitted that they overspend when using this tool.

BNPL companies claim that they are more responsible than credit cards because they are more inclusive, have fairer loan terms, and offer a convenient option for responsible consumers who pay back their debts on time (Aalders 2023).

A target market that BNPL companies are watching with increasing interest is the younger consumer segment, who although more financially unstable is at the same time more open to digital credit services (Coffey et al. 2023). These customers are interested in using BNPL options to keep up with fashion (fashion mavens) or to finance their social activities and hobbies (social recreationists). However, there are also people in older age groups who use BNPL for essential needs (conscious families) or even older people looking for bargains (hill chasers). A customer-first BNPL experience can be created by focusing on financial wellness, trust, and inclusion, delivered to the customer through an innovative BNPL portfolio and a holistic service experience.

Many customers are ready to expand the purchases that they make with BNPL. Studies have revealed that 42% are interested in using BNPL for future medium-sized purchases— 37% for large purchases (Deloitte Digital 2023). Online sellers increase their sales, and customers gain more control over their budgets and purchases. In June 2022, the US corporation Apple announced that it will enter BNPL market, dominated by start-ups such as Sweden's Klarna and the US-based Affirm. That same month, PayPal announced its own BNPL service, Pay Monthly (Oxford Business Group 2022).

In 2019, the $60B BNPL market represented 2.6% of global e-commerce (excluding China); Worldpay estimates that it could grow at a compound average growth rate of 28% to reach $166B by 2023, and digital wallets, the next most popular e-commerce type, are expected to grow at a compound average growth rate of 18% according to Consumer Banker (Consumer Banker Association 2022).

Last year, BNPL accounted for $2 out of every $100 spent in e-commerce, according to Global Data (GlobalData 2022). With a global market value of $125B in 2021, the segment is projected to exhibit a compound annual growth rate of 24.9% and reach $3.9T by 2030. In emerging markets, the share of adults making or receiving digital payments rose from 35% in 2014 to 57% in 2021, according to the World Bank (The World Bank 2022). According to Statista, at the global level, the BNPL market grew from 0.4% in 2014 to 5% in 2022 from domestic e-commerce. The most important increase was in Germany (from 3% in 2014 to 23% in 2022), and the smallest was in Canada, adopting BNPL only in 2021 (Statista 2023a).

Discussing the growth of the market, according to a report (Research and Markets 2023) the global buy now pay later market size is expected to reach $39.41B by 2030, registering an increase in the compound annual growth rate of 26.1% from 2022 to 2030. This growth is an effect of higher purchasing power determined by BNPL platforms and their perceived benefits, namely interest-free and convenient payments. Additionally, the increasing demand for delayed payments determined by online shopping and a surge in

spending on high-cost products determine will also be lucrative growth opportunities for the market by 2030.

By 2022, according to the same report, North America dominated the market with the highest share of around 29.0% of global revenue due to many BNPL players and a large number of fintech companies entering into partnerships with entertainment companies to offer BNPL services for booking hotels.

When discussing retail segment, it is the one that dominated the market in 2022 with more than 73.0% of the global revenue since the industry is characterised by increased adoption of BNPL solutions.

Additionally, the consumer electronics segment will register CAGR growth over the 2022–2030 period because the industry is witnessing increasing adoption of BNPL payment methods. Customers prefer BNPL payment methods to credit cards to avoid expensive interest and hidden fees. The report also stated that increasing awareness among consumer electronics providers about BNPL will determine the growth of demand for these services between 2022 and 2030.

In this context, the authors intended to analyse the market share in domestic e-commerce payments as a dependent variable. The argument for selecting this indicator is represented by rising adoption as a "key innovation in consumer finance in recent years" (Berg et al. 2023). The reasons for choosing logistic performance, ease of doing business, e-government development, postal development, and internet inclusivity and linking them to the market share of BNPL in domestic e-commerce payments, hereinafter MSDEP (%), determined the aim of creating a link with some inherent performance indicators related to the activity of developing electronic commerce.

The role of logistics performance in e-commerce is crucial due to the increase in B2C order volumes, with the literature presenting various opportunities, such as service outsourcing, to reach high levels and gain the capacity to deliver products quickly and cost-effectively (Pua 2023). The actual context presents challenges related to delivery guarantees (which are more indispensable for large companies than for small businesses) (Yang et al. 2023), product returns, and customer satisfaction (Saravanan et al. 2023). Representing an extremely important branch for the development of electronic commerce based on the adoption of the BNPL payment method, the authors hypothesise that:

**H₁.** *Logistics performance positively influences the market share of BNPL in domestic e-commerce payments.*

The ease of doing business is promoted and strongly related to digital business (Gupta 2022; Saidu et al. 2022), especially for the Fintech industry, for which the literature shows that using such instruments can ease e-commerce suppliers' capital (Park and Ryu 2023). The hypothesis related to this indicator is:

**H₂.** *Ease of doing business positively influences the market share of BNPL in domestic e-commerce payments.*

Regarding e-government development, it can be said that it represents a sustainable development method, especially for electronic commerce (MacLean and Titah 2022; Theocharis and Tsihrintzis 2023). E-government, information, and communication technology contribute to the development of e-commerce (Kastratović and Bjelić 2022), which is the why the authors expect the following:

**H₃.** *E-government development positively influences the market share of BNPL in domestic e-commerce payments.*

The postal service "is an indispensable part of electronic commerce" (Goyal and Morgan 2023), and its reliability plays a crucial role in development. It is expected that:

**H₄.** *Postal development positively influences the market share of BNPL domestic e-commerce payments.*

After all, internet inclusion plays an important role, representing the criteria for benchmarking adoption of e-commerce (Goyal and Morgan 2023). Reflecting on its connection with the BNPL method, the authors hypothesised:

**H₅.** *Internet inclusion positively influences the market share of BNPL domestic e-commerce payments.*

## 3. Methodology

The market size of BNPL is growing due to online payments that are rapidly growing (Bian et al. 2023), but a question may be raised regarding how ready e-commerce is to cover the demand. Attempting to identify some solutions and opportunities for companies and policy makers regarding this emergent technology, the authors addressed a mixed-method scenario. First, with high interest in understanding the status and context of the BNPL concept within the specialised literature, a bibliometric analysis was performed (Allen et al. 2009; Dhamija and Bag 2020). Second, the authors found and analysed secondary data about the market share of BNPL in domestic e-commerce payments (Statista 2023a) to describe the status of the phenomenon's spread. The findings are summarised within descriptive statistics analysis section. Finally, desk research (Borowski and Karlikowska 2023; Rahman et al. 2014; Yang 2023) was used to identify significant factors to explain whether the world is ready to buy now and pay later for products using the internet.

### 3.1. Bibliometric Analysis

In the first stage, the authors intended to place the BNPL concept in the literature using bibliometric analysis. As a scientific computer-assisted review method (Lazarides et al. 2023), it helped to identify the core significance of BNPL in science, also creating connections with other topics. Using specialised software, CiteSpace (2003), the authors attempted to find major areas related to BNPL and to connect the papers published on this topic by title, keyword, and abstract term nodes. By applying the clustering process, the authors labelled the major areas and interpreted the betweenness centrality score to explain how the major areas are linked. The validation of the network that they created was realised by interpreting the overall structural properties with modularity Q and mean silhouette indicators.

### 3.2. Empirical Model

Attempting to decode the information presented in the market growth sub-section from the literature review, the researchers selected BNPL market share in domestic e-commerce payments for 23 countries worldwide from Asia, Europe, the Middle East, North America, and South America from 2019 to 2022 to perform a descriptive statistical analysis. The countries were selected based on the public data available from Statista 2022, so secondary data were used. The year 2022 was considered because, for this year, the majority of the countries presented published data. The analysis provided basic information about BNPL concepts, and calculated and logically presented useful data about the variables. To present the information obtained, graphical and pictorial methods were used, created with Cognos IBM software (IBM 2023). Therefore, the set of simple linear regressions allowed the authors to analyse the influence of some potential influential factors on BNPL market share carefully and separately regarding domestic e-commerce payments. By addressing sustainable development goals, five endogenous factors were selected to explain the dependent variable (Retailx 2021).

**Logistics Performance Index (LPI)** (The World Bank 2023a) is a tool presented by the World Bank as an interactive benchmark for countries to analyse and optimise their logistics

performance. The indicator was used as the rank for each country included in the analysis to explain MSDEP with a simple linear regression equation:

$$MSDEP_i = \beta_0 + \beta_1 LP_i + \varepsilon_i \tag{1}$$

where

- $MSDEP_i$ represents the market share of BNPL in domestic e-commerce payments of country $i$; and
- $LP_i$ represents the logistics performance index of country $i$. The connection between market share and logistics performance is highly sustained by the literature, with some recent studies presenting the market share and e-commerce platform development from the perspective of logistics performance (Florido-Benítez 2023; Prajapati et al. 2023; Zhou et al. 2023). These facts led the authors to formulate the first hypothesis of the study, namely ($H_1$), to identify whether LPI influences MSDEP.

1. **Ease of doing business (EDB)** (The World Bank 2023b) is the indicator published by the World Bank, and it measures business regulations across world economies. The lower that the score is (the rank), the easier that it is to do business in a certain economy. Recent studies (Alaeddine 2023; Costantiello and Leogrande 2023; Gizaw et al. 2023) have used this indicator, and the authors intended to determine whether it might be a significant factor explaining MSDEP, which led to the second hypothesis ($H_2$). For this purpose, they formulated the equation:

$$MSDEP_i = \beta_0 + \beta_1 EDB_i + \varepsilon_i \tag{2}$$

where

- $MSDEP_i$ represents the market share of BNPL in domestic e-commerce payments of country $i$; and
- $EDB_i$ represents the ease of doing business indicator of country $i$;

2. **E-Government Development Index (eGD)** (United Nation 2023) is a value based on three dimensions, namely an online services index, a telecommunication infrastructure index, and a human capital index. The online service index shows the extent to which the government can provide online communication and support services to citizens. The telecommunication infrastructure index is the score that shows the capacity of a country in terms of ICT infrastructure. Finally, the human capital index represents the inherent human capital. As a methodology, it represents the weighted average of normalised scores of the three previously mentioned variables, and it is calculated as:

$$eGD = \frac{1}{3}(OSI_{normalized} + TII_{normalized} + HCI_{normalized}) \tag{3}$$

For this study, the authors used the ranks of the countries involved in the analysis as explanatory variables for MSDEP with simple linear regression:

$$MSDEP_i = \beta_0 + \beta_1 eGD_i + \varepsilon_i \tag{4}$$

where

- $MSDEP_i$ represents the market share of BNPL in domestic e-commerce payments of country $i$; and
- $eGD_i$ represents the e-government development index of country $i$;

The researchers expected to discover whether the eGD influences the MSDEP to a certain extent, presented as the third hypothesis ($H_3$).

3. **UPU'S Integrated Index for Postal Development (2IPD)** (Universal Postal Union 2023) is a metric that uses postal big data to highlight postal development around the world. Its rank is given depending on four dimensions: reliability, reach, relevance,

and resilience. It is very important for e-commerce development and useful for policy makers, regulators, and operators. Additionally, the scientific world has presented some papers addressing postal development as crucial factor (Kenyon et al. 2005; Mokgohloa et al. 2019). The authors hypothesised that postal development is a significant factor for the market share of BNPL in domestic e-commerce payments ($H_4$).

$$MSDEP_i = \beta_0 + \beta_1 2IPD_i + \varepsilon_i \tag{5}$$

where

- $MSDEP_i$ represents the market share of BNPL in domestic e-commerce payments of country $i$; and
- $2IPD_i$ represents UPU's integrated index for postal development of country $i$;

4. **Internet Inclusive Index (III)** (Economist Impact 2022) is a rank created based on scores for four categories: availability, affordability, relevance, and readiness. It was commissioned by Facebook and developed by the Economist Unit, and its aim is to enable outcomes for social and economic development. Hence, the authors used this indicator as an explanatory variable for MSDEP to determine whether the rank of the country for its perspective influences the market share of BNPL in domestic e-commerce payments for that country ($H_5$).

$$MSDEP_i = \beta_0 + \beta_1 2III_i + \varepsilon_i \tag{6}$$

where

- $MSDEP_i$ represents the market share of BNPL in domestic e-commerce payments of country $i$; and
- $III_i$ represents the internet inclusive index of country $i$;

## 4. Results

By performing bibliometric analysis, the clustering process showed 23 underlying themes related to BNPL line research. The most important of them are illustrated in Figure 2.

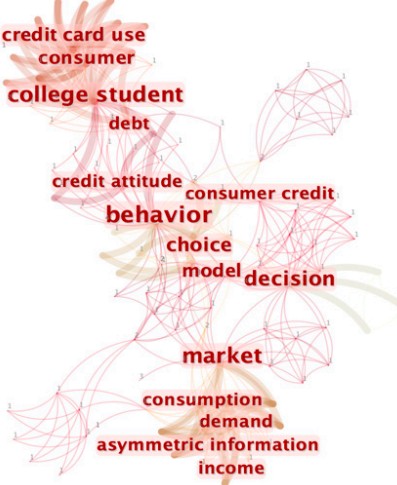

**Figure 2.** Results of clustering process. Source: created by authors with Citespace software.

The image shows that "credit cards", "behaviour", "market", and "college students" are relevant topics related to BNPL. Details of the top six largest clusters are available in Table 1. The column in which the year is shown highlights that cluster #11 has the most recent paper category with related concerns. After cluster label generation by title, keyword, and abstract term, a comparison was possible. It is shown in Figure 3.

**Table 1.** Details about generated clusters.

| Cluster ID | Size | Silhouette Mean | Year | Label (LSI) |
|:---:|:---:|:---:|:---:|:---:|
| 0 | 31 | 0.714 | 2014 | Credit card use; college students; hyperbolic discounting; experimental study |
| 2 | 21 | 0.913 | 2016 | Cross-border e-commerce; platform competition; internet-based financial innovation; conversion tools; consumer credit; impulse buying tendencies; conversion tools; consumer credit; cross-border e-commerce; platform competition |
| 3 | 20 | 0.826 | 2013 | Deferred payments; consumer decisions; mental accounting; dark financial triangles; consumer decisions |
| 4 | 16 | 0.881 | 2009 | Risk aversion; mental accounting; dark financial triangles; mental accounting |
| 11 | 7 | 1 | 2020 | Income tax credits; behavioural economics |
| 13 | 7 | 0.967 | 2019 | Adoption of e-wallets; perceived ease of use; perceived usefulness; perceived security; perceived benefits |

Source: created by authors based on Citespace outputs.

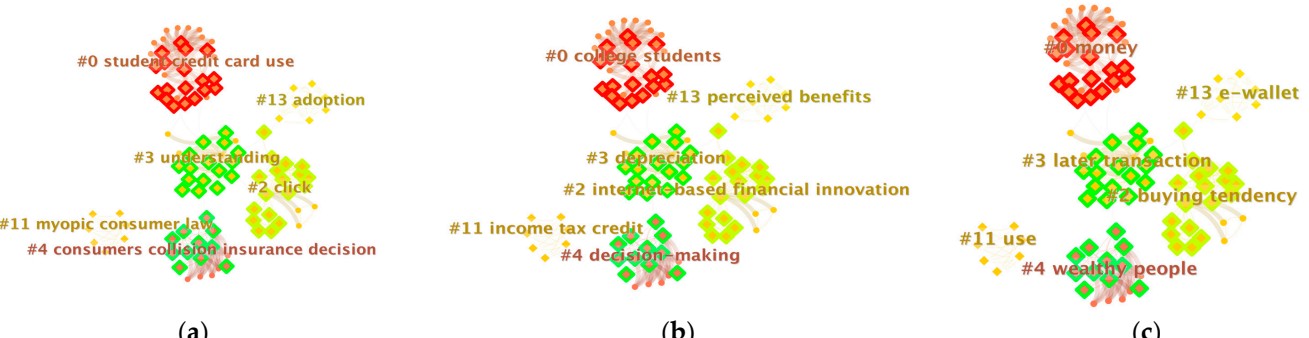

**Figure 3.** The cluster labels of the network. Extracted noun phrases from: (**a**) titles; (**b**) keywords; and (**c**) abstract terms.

By analysing the comparative image, it can be shown that different topics are highlighted according to titles, keywords, or abstract terms. Depending on titles, "student credit card use" represents the largest cluster, while "college students" was the largest cluster by keywords. According to abstract terms, "money" is the label with the largest cluster.

The main objective of using bibliometric analysis was to find an answer for the question "*What are the major areas?*", and the result is pictured in Figure 4.

The largest cluster is #college student, which here indicates the largest number of member references. The clusters #0, #2, #3, and #4 highlight the general overview of BNPL concepts during 2000 and 2023. Additionally, the authors found an answer for the question "*How are these areas connected?*". Using the betweenness centrality score [0, 1], they created the network with nodes that show this fact (Figure 5). A node with high betweenness centrality connects two or more large groups of nodes with the node itself (square shape).

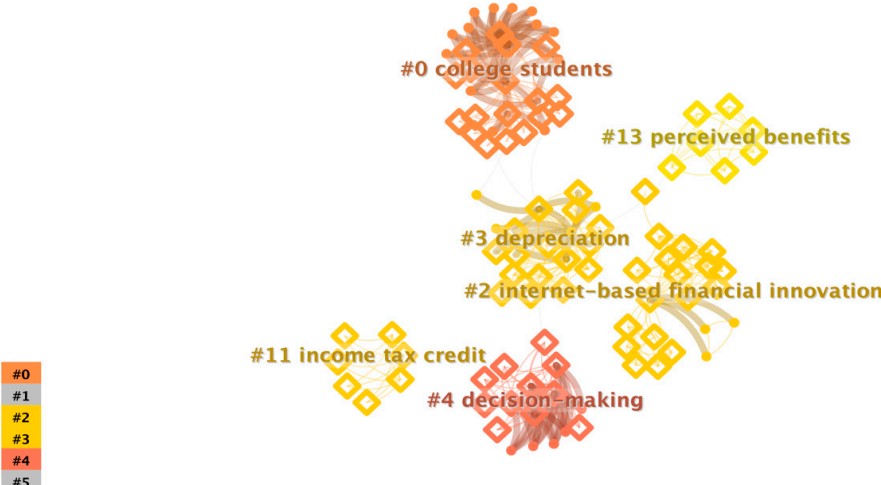

**Figure 4.** The major areas of the network. Source: created by authors with Citespace software.

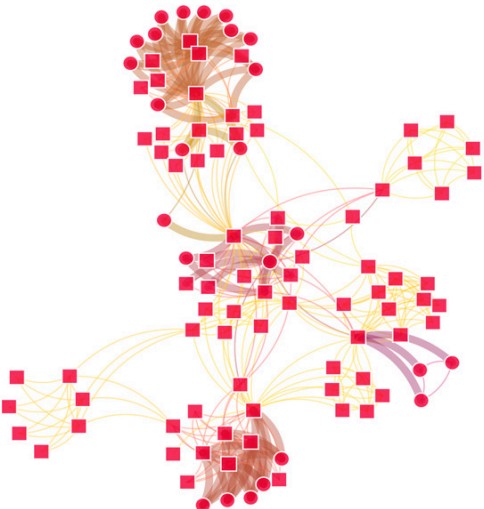

**Figure 5.** Major areas interconnected. Source: created by authors with Citespace software.

*BNPL Descriptive Statistics and Regression Analysis*

Figure 6 shows the market share of BNPL in domestic e-commerce payments registered in 2022 for several countries around the world, especially Germany and Australia.

The average of market share shows that, in Europe and Asia, the BNPL method has high potential (Figure 7), which can be analysed in the context of e-commerce development, showing high rates. Conversely, there are countries around the world that have not registered movements in terms of market share of this type of payment method (economies such as Peru, Brazil, Argentina, Turkey, and South Korea). Hence, major differences between countries are shown. More descriptive statistics are shown in Table 2.

The empirical model completes the analysis by covering the global context, with country analysis using exogenous factors and the structures of the scores, pictured in Figure 8 with stacked columns (aggregate scores for each country with the aim of presenting the rankings according to all five indicators). The bars illustrate the aggregate score, which compounds the ranks of the economies involved in the study for the explanatory variables. The different colours represent the ranks of each exogenous factor. For example, the best-performing economies (USA, UK, and Australia) are shown, but Japan presents better performance in terms of logistics that influence the market share.

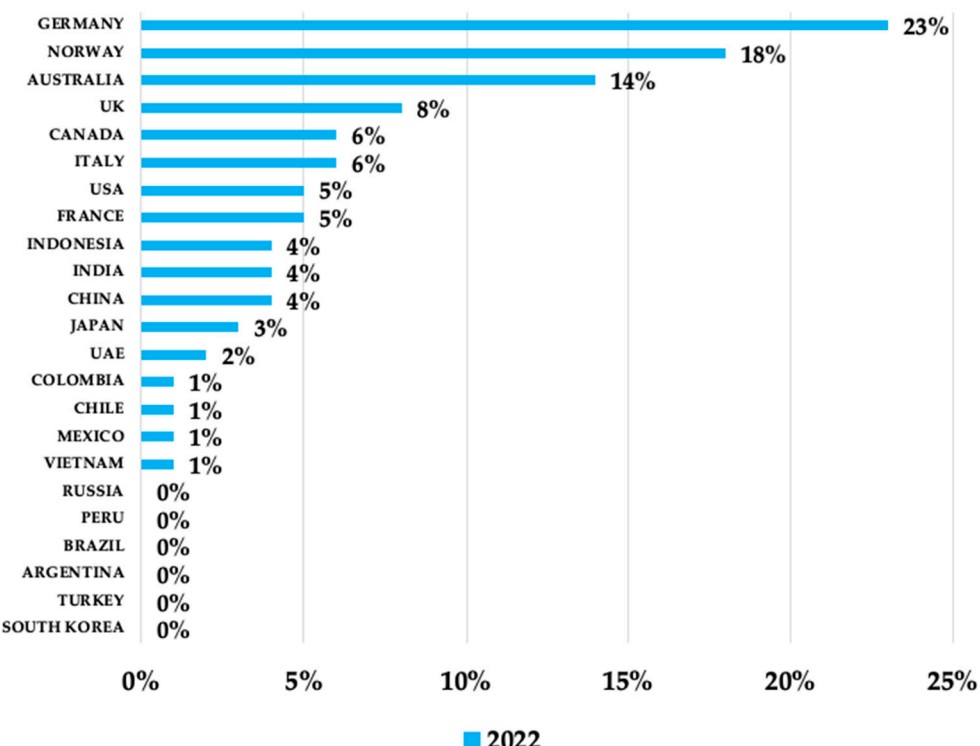

**Figure 6.** Comparison of BNPL market share in domestic e-commerce payments (2021 vs. 2022). Source: created by authors based on secondary data (Statista 2023b).

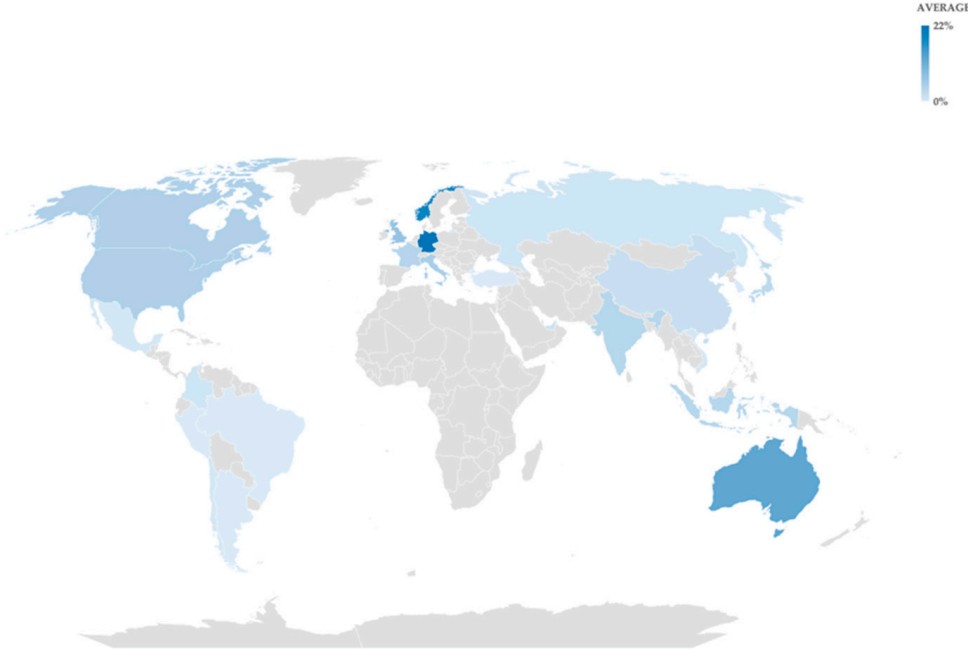

**Figure 7.** BNPL average market share in domestic e-commerce payments worldwide (2022). Source: created by authors based on secondary data (Statista 2023b).

**Table 2.** Descriptive statistics about MSDEP.

| Descriptive Statistics | MSDEP | LPI | EDB | eGD | 2IPD | III |
|---|---|---|---|---|---|---|
| Mean | 0.033 | 33.870 | 44.870 | 39.435 | 37.391 | 31.5 |
| Standard Error | 0.009 | 4.746 | 7.199 | 5.957 | 6.009 | 4.285 |
| Median | 0.018 | 26 | 32 | 34 | 34 | 30.5 |
| Mode | 0 | N/A | N/A | N/A | N/A | 25 |
| Standard Deviation | 0.044 | 22.760 | 34.526 | 28.567 | 28.817 | 20.099 |
| Sample Variance | 0.002 | 518.028 | 1192.028 | 816.075 | 830.431 | 403.976 |
| Kurtosis | 5.514 | −0.581 | 0.563 | −0.570 | −1.267 | −1.258 |
| Skewness | 2.247 | 0.547 | 0.984 | 0.631 | 0.418 | 0.132 |
| Range | 0.180 | 82 | 121 | 98 | 88 | 64 |
| Minimum | 0 | 1 | 5 | 2 | 3 | 2 |
| Maximum | 0.180 | 83 | 126 | 100 | 91 | 66 |

Source: created by authors based on secondary data.

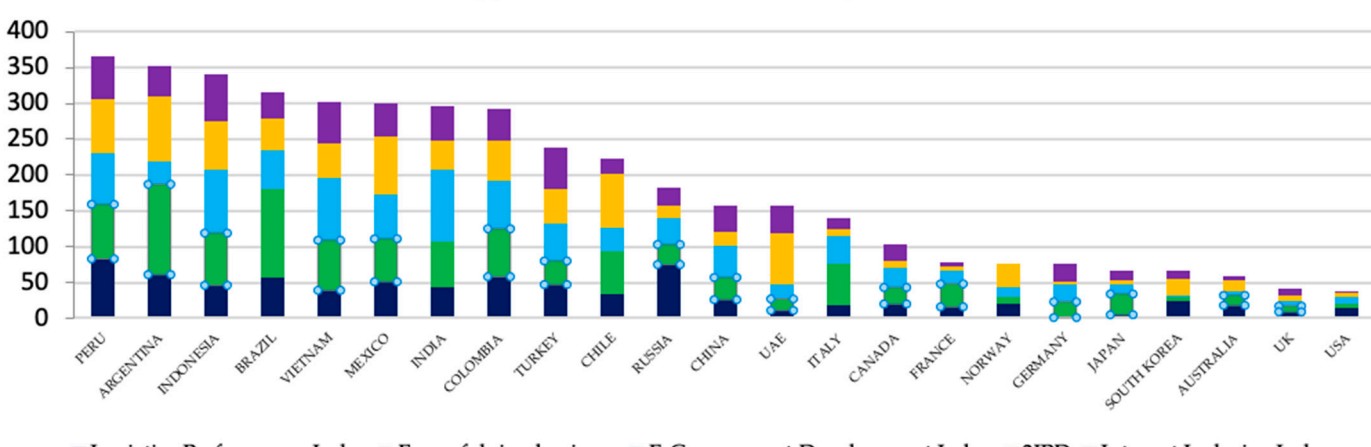

**Figure 8.** The structure of ranking by LPI, EDB, eGD, 2IPD, and III, (2022). Source: created by authors based on the secondary data (Economist Impact 2022; The World Bank 2023a, 2023b; United Nation 2023; Universal Postal Union 2023).

The empirical model analysis generated results regarding logistic performance index, ease of doing business, UPU's integrated index for postal development, and internet inclusive index that are significant variables, impacting the market share of BNPL in domestic e-commerce payments (Table A1). An unexpected result was found regarding the LPI (which moderately drives MSDEP). MSDEP is unusually high when LPI is 67 or greater, indicating that countries that present high ranks of LPI show a high market share of BNPL in e-commerce domestic payments. High scores of LPI do not highlight efficient and highest-ranked countries in terms of logistics performance, so in a paradoxical manner, these countries have a higher market share of BNPL in domestic e-commerce payments even though they do not perform as well in terms of logistics. LPI of 18 to <34 has the lowest average MSDEP at 1%, followed by the LPI group of 51 to <67 at 2%. This finding means that countries that perform better than the previous group have a lower BNPL market share.

Even in the case of the EDB variable, the facts are not as the authors expected. The highest average of MSDEP (at 9%) is given by the scores from 30 to <42, followed in second place by the group from 18 to <30 (6%). However, the highest frequency that appears for

this variable is for the less than 18 group (26.1%), which balances the results. Another anomaly is also found in the case of performance regarding the 2IPD variable because the group from 39 to <56 has the highest average of MSDEP at 7%, followed by 74 and above at 5%, once again emphasising that a large market share of BNPL in domestic e-commerce payments does not depend on the country's performance regarding 2IPD.

Additionally, the authors introduced Table 3, the correlation matrix (based on Pearson's coefficients used to identify patterns) to better visualise the relationship between indicators.

**Table 3.** Correlation matrix.

|  | **MSDEP** | **LPI** | **EDB** | **eGD** | **2IPD** | **III** |
|---|---|---|---|---|---|---|
| MSDEP | 1 |  |  |  |  |  |
| LPI | −0.549 * | 1 |  |  |  |  |
| EDB | −0.448 * | 0.659 * | 1 |  |  |  |
| eGD | −0.378 | 0.627 * | 0.621 * | 1 |  |  |
| 2IPD | −0.464 * | 0.64 * | 0.643 * | 0.524 * | 1 |  |
| III | −0.387 * | 0.641 * | 0.57 * | 0.859 * | 0.715 * | 1 |

Source: created by authors based on secondary data using Pearson's function. * Significant for 1%, 5%, or 0.1%.

Based on this table, one of the most important result of the study can be observed. Whereas all hypotheses are stated for positive influence, the correlation matrix showed negative values, leading to the refutation of all hypotheses.

## 5. Discussion

The dilemma that this article puts at the centre of the authors' concerns is finding an answer to the question "*Is the world ready to buy now and pay later?*" from an e-commerce perspective. Thus, the purpose of the research was to identify the factors that determine the market share of BNPL among domestic electronic commerce payments. Using data available for economies worldwide, the authors compared the ranks and obtained relevant outcomes for the knowledge field. For a better visualisation of what they have discovered, Figure 9 is provided.

Starting from the first objective (O1), namely describing the core significance of BNPL payment methods scientifically, it was found that the concept is described in the literature by popular factors, such as internet-based financial innovation, depreciation, and decision making. However, the results of the bibliometric analysis carried out in the first stage of the study reveal that core significance is not described as much from the point of view of electronic commerce, a result that further strengthens the results obtained by addressing the empirical model.

The second objective (O2) was achieved and presented "student card credit", "myopic consumer law", "decision making", "later transaction", "e-wallet", "buying tendency", and "insurance decision" as topics related to BNPL, enriching the outcomes of the study because they will be linked with the model results. In this context, the related topics that were found as results of the bibliometric analysis show a high risk in terms of perceived benefits (Gerrans et al. 2022; Relja et al. 2023). A young person who needs and wishes to buy as soon as possible a product available on e-commerce platforms but does not have the money to buy it, can access BNPL payment methods, and all the risks related to the found terms can occur. The outcome of the study confirms the concern of specialists regarding the distorted behaviour that BNPL is consolidating (Soni 2023), especially among Millennials (Arisandy et al. 2023; Min and Cheng 2023).

Therefore, by identifying the logistics performance index, ease of doing business, 2IPD, and internet inclusive index as determinant factors, the third objective (O3) was achieved. Thus, the results of the study support Gupta (2022) and Saidu et al. (2022), who promoted the ease of doing business as being strongly related to digital business, and Goyal and

Morgan (2023), who stated that the reliability of postal development is indispensable for e-commerce.

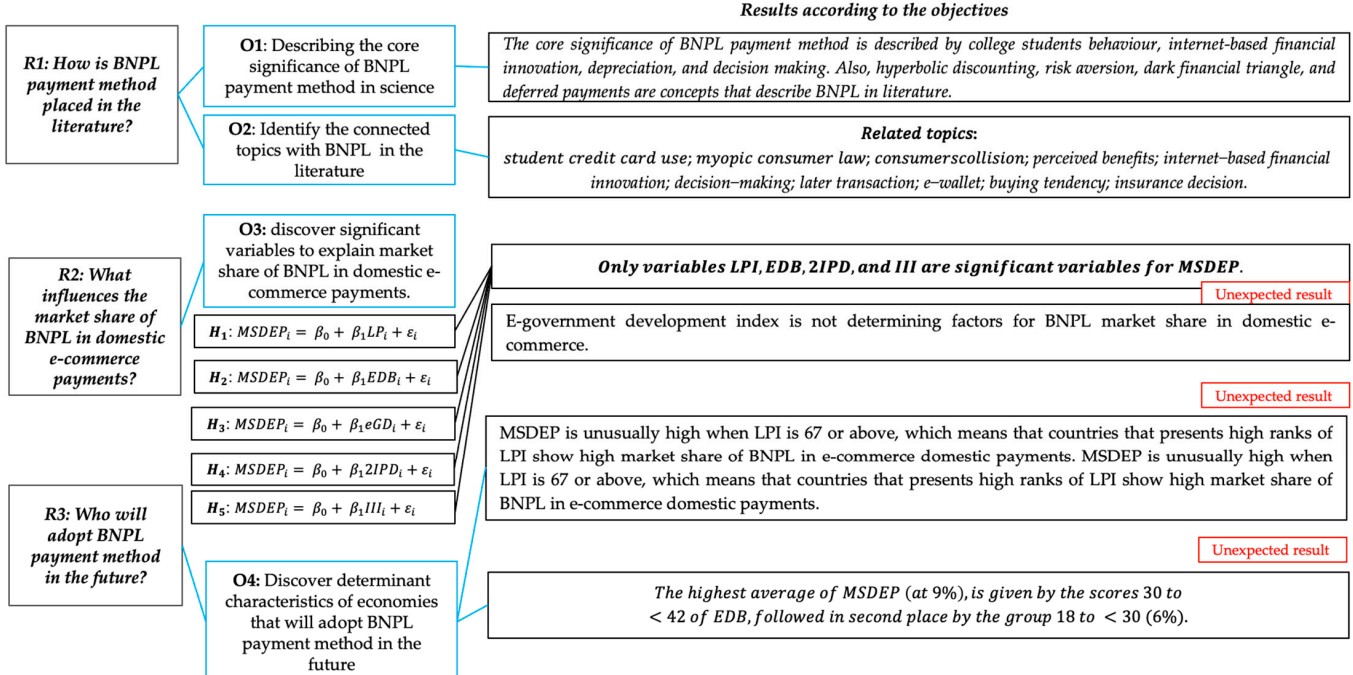

**Figure 9.** Results of the study according to research questions and objectives. Source: created by authors.

The fourth objective was achieved by discovering the determinant characteristics of economies that will adopt BNPL payment methods in the future. Compared to previous studies or published statistics, the paper responds to an important question: *Who will adopt BNPL payment methods?* In fact, a better and suitable question might be *"In which quartiles does an economy have to fit as a country depending on the factor to be prepared for BNPL?"* According to researchers, economies with a medium rank of ease of doing business present a large market share of this kind of payment method, and they are inclined to adopt it easily. This finding could lead to discussions that might affect the integrity of entrepreneurs when it comes to actions that they must undertake to run their businesses to resist that market. Moreover, the uncertainty that gravitates around BNPL payment method (Statista 2023b) might be linked to political instability and policymaker corruption (Albritton 2019). Additionally, this study comes with a macro-perspective on the economies predisposed to adopt this payment method more quickly or more easily, although they are not prepared in terms of inclusive internet performance or e-government development. Even in terms of postal development, it can be said that a large market share of BNPL in domestic e-commerce payments does not depend on the country's performance regarding 2IPD, indicating that economies with smaller market shares present better performance. This fact is interesting in the actual context. The accelerated development of technologies and applications (such as the Revolut) further deepen the gap between balanced economies from this point of view and those that currently experience increases in BNPL's market share of domestic e-commerce payments, especially since artificial intelligence is becoming an increasing part of everyday activities.

## 6. Conclusions

The scientific contribution of the study is that it enriches the specialised literature with a macro-perspective at an international level regarding the phenomenon of adopting the BNPL payment method. Additionally, the paper presents a perspective on the situation in

which economies are prepared or not to adopt BNPL as a payment method for domestic e-commerce.

Logistics development, ease of doing business, and e- and postal development are presented as aspects to be addressed with increased priority in countries where MSDEP grows annually and where consumers are becoming increasingly vulnerable. Additionally, increased attention be mentioned from policymakers regarding postal development, in which the market share of BNPL in domestic e-commerce payments is large. As the literature and this study demonstrated, this indicator might represent the key element for the sustainability and future development of e-commerce. Nor should topics popular in the specialised literature be neglected. Hence, major implications, both managerial and academic, must be addressed.

Managerial implications are complicated to address because, while BNPL can increase sales, the assumption of risk by companies or tech giants, which with the help of applications can accelerate the adoption process of this type of payment, is very important. The entities that might be involved should ensure transparency and responsible practices to sell products using this method. Additionally, companies can educate and protect consumers and monitor the behaviour of some groups considered more vulnerable, especially Millennials. A high responsibility should be borne by industry associations, which must run information campaigns and provide data to authorities and responsible institutions.

From the point of view of theoretical implications, studying the BNPL concept and its outcomes might deepen understanding of consumer behaviours, decision-making processes, risk perception mitigation, debt behaviours, and credit adoption.

A first limitation of the study is the lack of customisation of the manner in which the research was performed. Therefore, secondary data do not allow for the production of raw data for analysis, and potential bias might have occurred. Even if this type of research in general cannot be perfectly aligned with the objectives of the study, the authors intended to formulate objectives achievable with secondary data analysis. Because of the nature of the data source, the manner of choosing only five indicators, there being only 23 countries for which there existed data, and the analysis being limited only to the timespan between 2019 and 2022, other limitations of the study occurred. However, given the preliminary nature of research, this limitation can be overcome if the study continues, with its findings contributing to methodology improvement. Additionally, that the authors collected the data from different sources or references determined that they were less reliable than collecting them from a single source. This fact represents another limitation of the study, but despite these limitations, the researchers state that the outcome of the study is a valuable resource for creating further research lines and could provide reliable perspectives on how BNPL might evolve.

Further research might pursue psychological analysis to decode consumer decision-making and buying behaviours in the context of myopic consumer law or dark financial triangles. Additionally, qualitative research could be conducted to create solutions for BNPL digital tools to increase transparency and financial education.

**Author Contributions:** Conceptualization, D.A.L.-T., E.N., L.D., I.B.C., R.D.T. and G.B.; methodology, E.N. and G.B.; software, E.N. validation, D.A.L.-T., L.D. and R.D.T.; formal analysis, E.N. and L.D.; investigation, E.N., L.D. and R.D.T.; resources, D.A.L.-T., L.D. and I.B.C.; data curation, E.N.; writing—original draft preparation, D.A.L.-T., E.N., L.D., I.B.C., R.D.T. and G.B.; writing—review and editing, E.N., L.D., I.B.C., R.D.T. and G.B.; visualization, E.N.; supervision, G.B.; project administration, G.B.; funding acquisition, G.B. All authors have read and agreed to the published version of the manuscript.

**Funding:** The APC was funded by Transylvania University of Brașov.

**Informed Consent Statement:** Not applicable.

**Data Availability Statement:** Available data on request.

**Conflicts of Interest:** The authors declare no conflict of interest.

## Appendix A

**Table A1.** Linear regression results.

| Dependent Variable | Independent Variables | $\beta_0$ | $\beta_1$ | Multiple R | $R^2$ |
|---|---|---|---|---|---|
| MSDEP | LPI | 0.088 | −0.001 | 0.549 ** | 0.30 *** |
| | EDB | 0.074 | −0.0007 | 0.448 * | 0.2 |
| | eGD | 0.07 | −0.0007 | 0.378 | 0.14 |
| | 2IPD | 0.075 | −0.0009 | 0.464 * | 0.21 |
| | III | 0.08 | −0.001 | 0.387 * | 0.22 |

* $p < 0.05$; ** $p < 0.01$; *** $p < 0.001$.

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
