# Peer review of "Buy Now Pay Later—A Fad or a Reality? A Perspective on Electronic Commerce"

_economies, doi:10.3390/economies11080218_

Round 1

Reviewer 1 Report

improve with references from 2022 and 2023

write limitations of study

no

Author Response

Please, see the attached document.

Reviewer 2 Report

The topic of the article is highly relevant in the present time. However, I believe it needs to be improved for publication in Economies. Therefore, I would like to provide my suggestions:

Abstract

In line 16, an abbreviation BNPL (Buy now pay later) appears, which needs to be defined.

Literature Review

The hypotheses should be presented here in the literature review section. The hypotheses should indicate whether the influence is positive or negative (e.g., H1: LPI positively influences MSDEP). Therefore, a better literature review is necessary to theoretically support the formulated hypotheses.

Methodology

The methodology needs improvement so that any researcher can replicate the study. Here are some points to improve:

• Why were these 5 indicators chosen? How was the selection of the 23 countries made? What criteria were used? Which countries are being analyzed? Why limit the analysis to the years 2019 to 2022?

• Were the data on various indicators obtained from the references mentioned in the text? The fact that the data come from different sources may make them less reliable (a limitation of the study).

• The measurement of the MSDEP variable should be clarified.

• In all hypotheses, MSDEP is the dependent variable. Why not apply multiple linear regression or panel data regression instead of simple linear regression? Considering that you have longitudinal data (from 2019 to 2022) from 23 countries, panel data regression would be a more appropriate approach to investigate the influence of the 5 indicators on the MSDEP variable.

• Explain what the letter i represents in equations (1), (2), (4), (5), and (6) because, in this case, there should be two letters: i and t, where i identifies the country and t identifies the year.

• In line 274, reference is made to recent studies. Which studies are being referred to? Citations should be provided.

• What is the E-Government Development Index (eGD), and what are the 3 dimensions that compose it? Clarify these concepts.

• It is mentioned that indicator 2IPD ranges from 0 to 100, but what scale is used to measure the other indicators?

• There are two identical titles, one in the methodology and another in the results section: "BNPL descriptive statistics and regression analysis." In the methodology, it could be called, for example, "Empirical model."

Results

The results section also needs significant improvement. Try to write in a logical manner to help understand the analyses being conducted to present the various graphs and tables. Here are some suggestions:

• You start with "The clustering process showed...," but clarify what process you are referring to. Describe in the text the main results obtained in the figures and tables.

• In Figure 6, only 2 years (2021 and 2022) are analyzed, while the other years are not. Why?

• Which years does Figure 7 refer to?

• What scale is used in the graph of Figure 8? Does it refer to a specific year? Clarify how the results for the Figure 8 graph were obtained.

• Present some descriptive results about the MSDEP variable.

• How did the variables used to evolve over time?

• I suggest presenting a table with the correlation matrix between the various indicators and the MSDEP variable. This table will help visualize the relationship between the variables.

• Figure 9 suggests that simple linear regression is not the best option because the authors are using all variables in the same model. Therefore, panel data regression would be more appropriate. In this figure, which values are represented (are they betas?), and which ones are significant at 5%, 1%, and 0.1%?

• From line 359 to line 375, I believe you are fulfilling the objective (O4) of identifying determining characteristics of economies that will adopt the BNPL payment method in the future. Clarify the analysis you are conducting and the literature you relied on to establish the indicated classes.

Discussion

The discussion section is incomplete, only discussing the bibliometric analysis.

Conclusions

Improve the conclusions based on the results obtained.

Minor editing of English language required!

Author Response

Please, see the attached document.

Reviewer 3 Report

Although the topic is quite interesting, there are some issues that need to be improved.

- There are several language issues. Some of them are discussed in the quality of English section.

- The dedicated keywords should be reviewed to assign the proper choices. For instance, "logistic performance" may not be the main keyword of the topic.

- The first two paragraphs of the introduction are generic about Fintech. We should start by talking generally about BNPL, which is our topic here. Also, we should talk a little about Millennials and Generation Z.

- Figure 1 is not representing a research framework. It is the workflow of the study. As a reader, it is expected to see domains, input and output variables to present the layout of the study and the expected outcomes. Also, research framework should be based on rigid literature review or/and a scientific theory/framework.

- The first research question is not clear and answering it would be a bit vague.

- There is a coherence issue that is needed to be solved with the copy editor. For instance, the first 3 paragraphs in section 1.2 are scattered, and not linked to each other.

- The paragraph in line 189 has no relation to its section.

The manuscript needs to be reviewed by an English native editor to improve its quality. Some notices points are:

- The abstract started with the word "nowadays". It is not preferred to use it in prestigious journals.

- The first line of the abstract has the abbreviation "Gen Z", but the complete name wasn't introduced first. 

- The citation in line 88 has extra square bracket.

- There should be a comma after the citation in lines 91 & 229.

- Line 120 needs to be corrected. The word "beware" doesn't make sense here.

- Sometimes, the concepts are written in the complete form, like "generation z" in line 162, but it was written in its abbreviation before that!

Author Response

Please, see the attached document.

Round 2

Reviewer 2 Report

The article underwent significant modifications, which improved the paper. However, there are still aspects to be improved.

I would like to draw attention to Table 2, where the maximum and minimum values are swapped. In this table, since you are studying only the year 2022, you could present descriptive statistics for all the main variables of the study.

In Tables 2 and 3, the values should be rounded to only two or three decimal places.

Regarding Table 3, you should indicate which values are significant. Also, please specify the correlation coefficient you are using and why.

According to the matrix presented in Table 3, the correlations of the various indicators show a negative correlation coefficient with the variable MSDEP, which is not at all good. Please note that the study's hypotheses state positive influences, and that is not what is happening. This is a significant problem for the study.

Where are the results presented to test the formulated hypotheses? Please provide a table or a diagram with the main results of the applied linear regression! Since you described the different models in the methodology section, what are the values of betas zero and betas one in the various equations?

In Figure 10, the hypotheses have not been altered.

Minor editing of English language required

Author Response

Thank you very much!

Reviewer 3 Report

Although there is a good improvement, but there are lots of typos. Unfortunately, it was a main comment from the first review, but it seems that the manuscript hasn't been presented to a qualified English editor. For examples:

1- A space before the full stop in line 24.

2- Line 33 started with a comma!

3- Empty square brackets in lines 52 and 57.

When targeting a reputed journal (any journal not only this), it is better to double-check the language, because it could harm the overall decision of the review process. 

As mentioned above. Unfortunately, the discussed language issue wasn't considered sufficiently in this round.

Author Response

Thank you very much!

Round 3

Reviewer 3 Report

Thank you for the good work. Well done!